# Sex and Age Impact CD4+ T Cell Susceptibility to HIV In Vitro through Cell Activation Dynamics

**DOI:** 10.3390/cells12232689

**Published:** 2023-11-23

**Authors:** Ludivine Brandt, Paolo Angelino, Raquel Martinez, Sara Cristinelli, Angela Ciuffi

**Affiliations:** 1Institute of Microbiology, Lausanne University Hospital and University of Lausanne, 1010 Lausanne, Switzerland; ludivine.brandt@chuv.ch (L.B.);; 2Translational Data Science (TDS)-Facility, AGORA Cancer Research Center, Swiss Institute of Bioinformatics, 1015 Lausanne, Switzerland

**Keywords:** HIV-1, HIV susceptibility, immune activation, sex differences, age differences

## Abstract

Cellular composition and the responsiveness of the immune system evolve upon aging and are influenced by biological sex. CD4+ T cells from women living with HIV exhibit a decreased viral replication ex vivo compared to men’s. We, thus, hypothesized that these findings could be recapitulated in vitro and infected primary CD4+ T cells with HIV-based vectors pseudotyped with VSV-G or HIV envelopes. We used cells isolated from twenty donors to interrogate the effect of sex and age on permissiveness over a six-day activation kinetics. Our data identified an increased permissiveness to HIV between 24 and 72 h post-stimulation. Sex- and age-based analyses at these time points showed an increased susceptibility to HIV of the cells isolated from males and from donors over 50 years of age, respectively. A parallel assessment of surface markers’ expression revealed higher frequencies of activation marker CD69 and of immune checkpoint inhibitors (PD-1 and CTLA-4) in the cells from highly permissive donors. Furthermore, positive correlations were identified between the expression kinetics of CD69, PD-1 and CTLA-4 and HIV expression kinetics. The cell population heterogeneity was assessed using a single-cell RNA-Seq analysis and no cell subtype enrichment was identified according to sex. Finally, transcriptomic analyses further highlighted the role of activation in those differences with enriched activation and cell cycle gene sets in male and older female cells. Altogether, this study brought further evidence about the individual features affecting HIV replication at the cellular level and should be considered in latency reactivation studies for an HIV cure.

## 1. Introduction

A virus’ success in establishing infection depends on the following two main cellular features: (i) the expression of surface receptors for viral entry and (ii) a cellular permissive environment favoring viral replication. In the case of the human immunodeficiency virus type 1 (HIV-1, hereafter abbreviated HIV), susceptible cellular populations have to express the CD4 entry receptor and the CCR5 or CXCR4 coreceptors, making CD4+ T cells the main target for HIV. The majority of these cells display a restrictive intracellular environment, making them initially refractory and non-permissive to viral infection, unless exposed to specific activation stimuli leading to cell population differentiation, hence increasing the heterogeneity at a cellular level. Consequently, the changing intracellular composition results in a heterogeneous response to viral infection. The diversity of CD4+ T cells is fueled by the following: (i) activation status, (ii) cell subset, and (iii) differentiation stage. Multiple studies aimed at identifying molecular determinants associated with increased permissiveness to HIV. Examples include cell types and features such as an enhanced CCR5 expression [1], all types of memory CD4+ T cells, especially effector memory cells, T helper (Th) 17 and T follicular helper (Tfh) subtypes [2,3], and cells displaying an activated or exhausted phenotype displaying a low type I interferon response [4].

Besides cellular composition, HIV replication success can be influenced by individual-specific features such as sex or age. Indeed, growing amounts of evidence show that sex can shape the immune response of an individual [5,6,7]. Sex-related differences can be explained with differential steroid hormone regulation, as well as genetic factors [5]. The latest epidemiological statistics reveal that women represent 54% of people with HIV (PWH) and are more susceptible to HIV seroconversion through heterosexual intercourse by a two-fold factor compared to men [8,9]. Although both sexes experience similar AIDS progression rates [10], women progress with initially lower viral loads [11]. Moreover, Scully et al. showed that HIV replication is less efficient in women under antiretroviral therapy (ART) upon a measurement of the viral transcriptional activity and the reactivation potential [12]. This observed lower replication may be the result of plasmacytoid dendritic cells in women that produce more type I interferon in response to HIV ligand-binding to TLR7, enhancing CD8+ T cell activation, provoking a higher expression of interferon-stimulated genes in CD4+ and CD8+ T cells and in dendritic cells from untreated women compared to the levels observed in men [13,14]. Last, estrogen receptor was shown to be a strong regulator of HIV latency: indeed, the estrogen binding to its receptor can repress viral reactivation, while the receptor’s blockade can promote it, outlining a key role of sex hormones in HIV biology [15]. Recently, Gianella et al. outlined that women post-menopause undergo a higher provirus reactivation compared to younger women, which is possibly caused by an estrogen decline [16]. Although a significant part of these sex-specific differences can be attributed to hormonal regulation, evidence demonstrates lower viral loads in prepubescent women compared to men of the same age, suggesting the involvement of other factors, such as genetic differences due to incomplete X inactivation [17,18].

Age is another major contributor to immune differences, with the first alterations being visible at the age of 50, with thymic involution leading to a reduced T cell generation [19]. With age, naïve/memory cell balance shifts towards increased memory cell frequencies, which is more pronounced in late-stage differentiation memory subtypes, displaying an enhanced effector function and a reduced proliferation capacity. In the HIV field, most studies address the effect of aging with HIV in order to improve older patients’ care, without investigating a potential direct impact of HIV on the cells of older individuals [20].

Here, we addressed whether these sex-based differences could be recapitulated in vitro, without hormonal influence and biases caused by a prolonged HIV infection. In addition, we interrogated the effect of age on cellular permissiveness to HIV. To this aim, we isolated CD4+ T cells from twenty HIV-negative individuals, stimulated them with TCR-mediated αCD3/CD28 over a period of six days, and infected them every day with three HIV-based vectors pseudotyped with VSV-G, CCR5- (R5-), or CXCR4- (X4-) tropic envelopes and harboring two fluorescent reporters, one LTR-driven GFP reporter and one constitutive EF1α-driven mKO2 reporter (HIV GKO [21]), which were monitored using flow cytometry. We compared a virus-encoded reporter, first between all the donors and then by sex and age, to reflect the permissiveness over time post-stimulation (p.-s.). We found that the viral infection peaked at 24 h–72 h p.-s., as expected. We then focused on this time window and found that the cells derived from men and individuals older than 50 years of age displayed an enhanced cell susceptibility to HIV, correlating with an increased expression of activation markers and immune checkpoint inhibitors (ICIs). A transcriptomic analysis revealed an enrichment of the activation and cell cycle gene sets in the male and aging individuals’ cells, supporting a major role of activation in the permissiveness phenotype. Finally, we identified a differential regulation in sex-linked genes that may impact the permissiveness to HIV.

## 2. Materials and Methods

### 2.1. Ethics Statement

All the blood donors provided written informed consent and all the samples were anonymized.

### 2.2. Cell Samples, Isolation, and Culture

Peripheral blood mononuclear cells (PBMCs) from HIV-negative blood donors were purified from whole blood samples through Ficoll gradient separation, using Leucosep tubes (Greiner, Kremsmünster, Austria) according to the manufacturer’s recommendations. Following purification, the PBMCs were frozen in heat-inactivated fetal bovine serum (HI-FBS) with 7.5% of dimethyl sulfoxide (DMSO, Merck, Darmstadt, Germany) in liquid nitrogen in cryotubes (Thermo Fisher Scientific, Waltham, MA, USA) for long-term storage.

Primary CD4+ T cells were isolated from the PBMCs through negative selection and magnetic separation using an EasySep Human CD4+ T Cell Isolation Kit (Stemcell Technologies, Vancouver, BC, Canada) according to the manufacturer’s instructions. They were cultured at a concentration of 10^6^ cells/mL in RPMI-1640 (Thermo Fisher Scientific), supplemented with 10% of HI-FBS (Thermo Fisher Scientific) and 50 μg/mL of gentamicin (Thermo Fisher Scientific) at 37 °C, with 5% CO_2_. One day after purification, the CD4+ T cells were stimulated in a T25 flask for four days by adding 25 μL/mL of ImmunoCult Human CD3/CD28 T Cell Activator (Stemcell Technologies) in a medium supplemented with IL-2 [200 IU/mL] (R&D Systems, Minneapolis, MN, USA). After four days, the culture medium was replaced with a fresh medium supplemented with IL-2 [200 IU/mL], and the cells were cultured for two more days. Cell viability was checked prior to each manipulation using 0.4% of trypan blue (Thermo Fisher Scientific).

### 2.3. Virus Production and Infection

HIV-based lentiviruses LTR-HIV-∆-env-nefATG-csGFP-EF-1α-mKO2 (referred to as HIV GKO) were produced through the co-transfection of 5 million HEK293T per 10 cm dish with 7.5 µg of HIV GKO (gift from Eric Verdin, Addgene #112234 [21]) and 2.5 µg of one envelope plasmid using jetPRIME transfection reagent (Polyplus-transfection, Illkirch, France) according to the manufacturer’s recommendations. The culture medium was replaced with a fresh 293 Serum-Free Medium III (Thermo Fisher Scientific) supplemented with glutaMAX (Thermo Fisher Scientific) 8 h post-transfection. The viral particles were collected 48 h post-transfection and filtered using a 0.22 µm filter (Merck). The viral titers were measured with a p24 immunoassay using an INNOTEST HIV Antigen mAb (Fujirebio, Tokyo, Japan) according to the manufacturer’s instructions.

Alternative envelopes were used as follows: the VSV-G encoded by pMD2.G plasmid (gift from Didier Trono, Addgene #12259 [22]), the X4-tropic LAI envelope encoded by pCI-X4 plasmid (gift from Robert Siliciano [23]), or the R5-tropic BaL envelope cloned in the pCl-X4 backbone. For this latter construct, the LAI *env* sequence in pCI-X4 was substituted with the one from the pNL4-3-BaL *env* [24] through KpnI-BlpI (New England Biolabs, Ipswich, MA, USA) restriction cloning. Single-reporter HIV-based lentiviruses for flow cytometry compensation were constructed through SgrAI-XmaI and BlpI-HpaI (New England Biolabs) restriction cloning for the deletion of GFP and mKO2, respectively.

Cell permissiveness to HIV was monitored through the GFP and mKO2 expression of the primary CD4+ T cells infected at 0 h, 24 h, 48 h, 72 h, 96 h, 120 h, and 144 h p.-s. and was assessed using flow cytometry 47 h p.-i. Additionally, the cells were infected 24 h p.-s. and further monitored using flow cytometry at 23 h, 47 h, 71 h, 95 h, 119 h, and 143 h p.-i. Briefly, this was performed by exposing 100,000 cells to 30 ng p24 equivalent of HIV GKO/VSV-G, 100 ng of HIV GKO/BaL and HIV GKO/LAI, or a mock treatment, in a volume of 110 µL in a 96-well U bottom plate. The infections were carried out in the presence of 4 µg/mL of polybrene (Merck) and spinoculation (1500 g, 90 min, 25 °C). After that, the cells were washed and resuspended at 10^6^ cells/mL in R10 supplemented with IL-2 [200 IU/L] and incubated for 47 h. The cells were fixed in 200 µL CellFix 1× (Becton Dickinson, Franklin Lakes, NJ, USA) to monitor the GFP and mKO2 expression.

### 2.4. Cell Surface Marker Staining

Cell surface marker expression was assessed using single fluorophore-conjugated antibody staining on 50,000 primary CD4+ T cells at 0 h, 24 h, 48 h, 72 h, 96 h, 120 h, and 144 h p.-s. The antibodies were purchased from Biolegend (San Diego, CA, USA) and aimed at measuring the marker expression involved in HIV entry (CD4, CXCR4, CCR5), cell activation (CD69, CD25, HLA-DR), and immune checkpoint inhibition (PD-1, CTLA-4, TIM-3) (Appendix A). Briefly, cells (50,000) were washed once using a FACS buffer (phosphate-buffered saline, PBS, Bichsel, Interlaken, Switzerland), 5% of HI-FBS, and 2 mM of Ethylenediaminetetraacetic acid (EDTA, Thermo Fisher Scientific)) and were incubated with 0.5 µL of each antibody in a final volume of 100 µL for 30 min at 4 °C. The cells were then washed with FACS buffer and fixed in 150 µL of CellFix 1×. Marker expression was assessed using flow cytometry.

### 2.5. Flow Cytometry

The flow cytometry analyses of the infected samples and the antibody-stained samples were performed using a Gallios machine on 10,000 events (Beckman Coulter, Brea, CA, USA; Flow Cytometry Facility, University of Lausanne). The infection success was measured on channels one (GFP) and two (mKO2) and the surface marker expression on channels two (PE) and six (APC). All flow cytometry graphs and analyses were generated using the FlowJo (v.10.7.1) software (Becton Dickinson). The dead cells were pre-determined using a Zombie NIR Fixable Viability Kit (Biolegend) at all experimental time points and were excluded from the gating strategy. The gating strategy was set using mock-infected and non-stained controls. The same gating strategy was used at each time point for all the donors and was set in order to minimize the background of false positive events. This background was then subtracted from the percentage of positive cells in the infected or stained sample.

### 2.6. Statistical Analyses

Statistical analyses and graphical distributions were performed using the GraphPad Prism (v.9.1.0) software (GraphPad Software, La Jolla, CA, USA). Kinetic comparisons were performed using two-way analysis of variance (ANOVA) or alternatively, parametric paired *t*-test. The area under curve (AUC) measurements were compared using ANOVA with a false discovery rate (FDR) method of Benjamini and Hochberg. The correlation analyses were made based on linear regressions.

### 2.7. Single-Cell RNA-Seq Library Preparation and Sequencing

The cells resuspended in PBS supplemented with 0.04% of bovine serum albumin (BSA, Merck) were loaded into a Chromium Next GEM chip K with a target capture of 10,000 cells per sample. Gene expression (GEX) libraries were generated using a Chromium Next GEM Single Cell 5′ Reagent Kit v2 (10× Genomics, Pleasanton, CA, USA) according to the manufacturer’s recommendations and were then sequenced using NovaSeq 6000 (Illumina, San Diego, CA, USA) through a paired-end 100 nucleotides dual-indexing protocol in the Lausanne Genomic Technologies Facility (LGTF).

### 2.8. Single-Cell RNA-Seq Analysis

The matrixes of the GEX reads were generated through alignment and read count using CellRanger (10× Genomics, v.7.1). They were then loaded into R (v.4.2.0) and analyzed using the Seurat package (v4) [25]. Briefly, the samples were filtered for poor quality (expressing less than 200 transcripts and displaying more than 15% mitochondrial genes). The samples were then normalized using SCTransform and mapped against reference for the generation of uniform manifold approximation and projection (UMAP) visualization and determination of subtype composition. The cell types were annotated with Seurat multimodal reference mapping (reference data set [25]) and with the R package singleR [26]. DICE [27] and Monaco [28] were used as references annotations.

### 2.9. Population RNA-Seq Library Preparation and Sequencing

The total RNA was extracted from 500,000 cells stimulated for 24 h or 72 h using a Quick-RNA Miniprep Kit (Zymo Research, Irvine, CA, USA) according to the manufacturer’s recommendations. mRNA libraries were prepared using Illumina stranded mRNA protocol with polyA selection and were sequenced using NovaSeq 6000 through a single-read 100 nucleotide protocol in the Lausanne Genomic Technologies Facility.

### 2.10. Population RNA-Seq Analysis

Quality control, trimming of quality sequences and sequencing adaptors, and read alignment were performed using nf-core/rnaseq (v.3.12.0) from the nf-core workflow collection [29]. The pipeline was executed with Nextflow (v.23.04.1) [30]. The counts and transcripts per million were estimated using Salmon (v.1.5.2) [31] and the human reference genome GRCh38. Lowly expressed genes, with an average expression per condition of less than one count per million, were removed from all the conditions with the filtered.data function from the NOIseq R package (v.2.36.0) [32]. Differential gene expression analyses were performed using DESeq2, edgeR, and limma and used threshold of *p* < 0.01 and Log2 fold-change > 0.5 [33,34,35]. The *p*-values were adjusted for multiple comparisons using the Benjamini and Hochberg method. Differentially expressed genes were considered when identified as such using the three algorithms. The *p*-value represented is the largest observed. The GSEA was performed with the clusterProfiler R package (v.4.2.2) with the following parameters: minGSSize = 10, maxGSSize = 1000, eps = 0, and *p*-value Cutoff = 0.05 [36,37]. The pathways with an adjusted *p*-value < 0.05 were deemed as significantly enriched. The hallmark gene set for the GSEA was obtained from MSigDB (v.7.5.1). The gene sets signature scores were computed using the GSVA R package (v.1.44.5) [38], and the signature scores for each donor and for significantly enriched pathways were plotted as heatmaps.

## 3. Results

### 3.1. Cell Permissiveness to HIV Evolves with Cell Activation Dynamics

To explore individual permissiveness to HIV in CD4+ T cells, we studied a cohort of 20 HIV-negative blood donors varying in sex and age (22 to 72 years old) through TCR-induced activation dynamics (Figure 1A). Resting CD4+ T cells from each donor were stimulated for a total of six days (144 h) (Figure 1B). Every 24 h, the cells were collected, and we assessed selected surface marker expression (Appendix A) and HIV infection as reported through HIV GKO (Appendix A) using flow cytometry. The cells were counted, and their viability was assessed before each manipulation. The average viability per donor per time point was always ≥90%. Our results showed that the cell proportion expressing HIV entry markers was stable over time p.-s., and it was ubiquitous for CD4 and CXCR4, but restricted for CCR5 (present only in 12% of the CD4+ T cells on average) (Appendix A). However, the relative expression of these markers, reflected by their mean fluorescence intensity (MFI), outlined increasing levels of CD4 but decreasing levels of CXCR4 and CCR5 over the stimulation period (Appendix A). In parallel, the expression of cell surface proteins related to activation, i.e., CD69, CD25, and HLA-DR, as well as ICIs, i.e., PD-1, CTLA-4, and TIM-3, were measured to monitor the cell activation status. As expected, the activation markers’ expression increased with time p.-s. with differential kinetics. The early activation marker CD69 increased and peaked at 72 h p.-s., with an average of 74% positive cells before decreasing, while CD25 was progressively upregulated over time upon TCR-mediated stimulation, until being expressed on the surface of most cells (93% on average at 144 h) (Appendix A). The late activation marker HLA-DR showed progressive upregulation, reaching 47% of HLA-DR+ cells at 144 h on average. The relative MFI of the activation markers reached a peak at 24 h p.-s. for CD69, before a steep decline (Appendix A). CD25 levels peaked at 96 h p.-s. and HLA-DR increased progressively over time. Following activation kinetics, ICIs are upregulated simultaneously with activation markers in order to constrain cell proliferation, as a control feedback loop. The three selected ICIs displayed increased expression over time, but with different proportions, as follows: the PD-1-expressing cells peaked at 120 h with 81%, the CTLA-4+ cells reached 45% at 96 h, and the TIM-3+ cells peaked with 78% at 144 h (Appendix A). Consistently, the expression levels of the three markers increased with stimulation and decreased in the late time points, at 96 h for PD-1 and CTLA-4, and at 144 h for TIM-3 (Appendix A).

Infection was carried out with a HIV GKO (HIV_LTR-GFP-EF1α-mKO2) dual-reporter vector pseudotyped with three different envelopes, as follows: VSV-G for amphotropic viral entry, and natural HIV envelopes LAI for CXCR4 tropism, and BaL for CCR5 tropism. The expression level of GFP and mKO2 was assessed 47 h post-infection (p.-i.) in order to discriminate productive (GFP+) from latent (mKO2+ GFP−) infection (Appendix A). HIV infection success was evaluated using three analyses in order to assess (i) the susceptibility, (ii) the population permissiveness, and (iii) the intracellular permissiveness (Figure 1C–E). The proportion of susceptible cells to HIV was assessed with the number of cells expressing the constitutive EF-1α-driven mKO2 reporter. The data revealed an increasing proportion of susceptible cells over time p.-s., reaching the peak at 72 h for the HIV GKO/VSV-G (26.4%) and the LAI (8.6%), and between 24 h and 48 h p.-s. for the HIV GKO/BaL (1.1%) (Figure 1C). The analysis of the relative GFP MFI reflected the level of LTR-driven GFP production in the total population and, thus, assessed the global cell permissiveness to HIV, i.e., the combination of cell susceptibility to HIV (from entry to integration) and successful viral expression (Figure 1D). The data mirrored cell susceptibility kinetics, suggesting that susceptible cells are similarly permissive, i.e., enabling productive HIV infection. Finally, in order to discriminate the impact of entry from the determinants of intracellular permissiveness, we investigated the GFP expression of the productively infected population (GFP+) and observed higher amounts of GFP expression at 24 h p.-s., suggesting that the intracellular environment favoring permissive infection is rapidly established upon TCR-mediated stimulation (Figure 1E). The data also showed a progressive decline of LTR-driven GFP expression over time p.-s., which could not be explained by the direct establishment of latent infection, as the proportion of mKO2+ GFP− infected cells was stable or declining over time (Appendix A). In addition, the stability of the LTR promoter in driving GFP expression over time p.-i. was assessed by infecting cells 24 h p.-s. and monitoring the fluorescent reporter’s expression at 23 h, 47 h, 71 h, 95 h, 119 h, and 143 h p.-i. (Appendix A). In this context, the GFP expression accumulated to peak at 47 h p.-i. and then tended to decline with time, in particular with HIV envelopes, likely due to the death of the infected cells. Indeed, the proportion of latent cells only weakly correlated with time (linear regression, HIV GKO/VSV-G, *, *p* = 0.03, R-square = 0.04; HIV GKO/LAI, *, *p* = 0.02, R-square = 0.04; HIV GKO/BaL, *, *p* = 0.03, R-square = 0.04), suggesting that latency establishment may not increase with time, but that the decline in GFP+ cells may rather be due to cell death.

As expected, our data showed that the CD4+ T cells’ permissiveness to HIV was higher upon TCR-mediated stimulation, in particular between 24 and 72 h p.-s. Despite the permissiveness differences among the donor cells, similar kinetics were overall observed.

### 3.2. Sex and Age Impact CD4+ T Cell Susceptibility to HIV Infection

To study the impact of specific features such as sex and age on CD4+ T cell permissiveness to HIV, the data were further analyzed according to these parameters. A comparison of HIV infection kinetics between women and men cells revealed that infection kinetics were similar between both sexes, but that men harbored increased numbers of susceptible cells compared to women between 24 and 72 h p.-s., in the time window where cells had been previously identified as being most permissive (Figure 2A). This difference was significant for both the VSV-G- and LAI-mediated entries (HIV GKO/VSV-G ****, *p* < 0.0001; HIV GKO/LAI **, *p* = 0.004), but not for the BaL-mediated entry (*p* = 0.91), which may be the result of the low level of infection success. Permissiveness as reflected through GFP expression revealed increased levels in men cells and showed significance for VSV-G-mediated entry and a trend for LAI-mediated entry (HIV GKO/VSV-G, * *p* = 0.01; HIV GKO/LAI, *p* = 0.08; HIV GKO/BaL, *p* = 0.34) (Appendix A). However, the intracellular GFP levels were similar between the sexes (HIV GKO/VSV-G, *p* = 0.74; HIV GKO/LAI, *p* = 0.43; HIV GKO/BaL, *p* = 0.44) (Appendix A).

The impact of age on cell susceptibility to HIV was evaluated by analyzing the proportion of susceptible cells (mKO2+) in the younger (<50 years old) and older donors (≥50 years old) (Figure 2B). As for the sex impact’s analysis, the kinetics of susceptibility displayed by each age category were comparable over time. However, when focusing on the 24–72 h window p.-s., the older donors exhibited an increased susceptibility to infection compared to the younger ones for the LAI (X4)-tropic virus (***, *p* = 0.0001) and for the BaL (R5)-tropic virus (*, *p* = 0.02). However, no impact was visible for the viruses entering via the VSV-G route (*p* = 0.70). Consistently, the permissiveness levels were increased upon LAI (X4)-tropic infection (**, *p* = 0.004) or VSV-G -mediated entry (*p* = 0.30) (Appendix A). The BaL (R5)-tropic virus only showed a non-significant trend (*p* = 0.06). The intracellular GFP levels were similar between the younger and older donors with each virus (HIV GKO/VSV-G, *p* = 0.10; HIV GKO/LAI, *p* = 0.48; HIV GKO/BaL, *p* = 0.66) (Appendix A). Similar results were obtained when age was treated as a continuous variable for the VSV-G- and LAI (X4)-mediated entries but not with the BaL (R5)-mediated route (Appendix A). For both sex and age impact, our results suggest that the difference occurs during entry, integration, and productive LTR-driven GFP expression. The data were further dissected to interrogate the impact of a combination of sex and age, looking at the area under the curve (AUC) displayed by susceptible cells during the relevant 24 h–72 h p.-s. window (Figure 2C). We could recapitulate the previous results, i.e., male cells displaying an increased susceptibility to VSV-G-mediated HIV entry compared to female cells (**, *p* = 0.004), and the cells isolated from older donors showing an enhanced susceptibility to LAI (X4)-mediated HIV entry (*, *p* = 0.03). The analysis of HIV GKO/LAI (X4) infection interestingly showed that, although male cells only tended to display an enhanced susceptibility to viral infection compared to female cells (*p* = 0.07), the combination of sex and age revealed a gradual susceptibility to HIV GKO/LAI infection with the lowest AUC displayed by younger women and the highest by older men (*, *p* = 0.02). This effect could not be observed when using HIV GKO/BaL (R5), potentially because of the low infection success rate in this setting.

The effect of sex and age was investigated similarly, on the latency establishment, by assessing the AUC corresponding to the proportion of latently infected cells (mKO2+ GFP−) over the total infected cells (mKO2+) in the 24 h–72 h p.-s. window (Figure 2D). Although sex did not appear to affect the latent infection mediated through VSV-G entry, the data suggest that age impacted the proportion of latently infected cells, with the younger donors displaying a higher AUC upon VSV-G- and LAI-mediated HIV entry (*, *p* = 0.003 and *, *p* = 0.002, respectively), but not upon BaL-mediated HIV entry (*p* = 0.65).

Sex and age’s impact on GFP expression stability was also controlled in p.-i. settings. No statistical difference between male and female cells was detected in GFP production over the total time of infection, with all three vectors, using a two-way ANOVA (HIV GKO/VSV-G, *p* = 0.28; HIV GKO/LAI, *p* = 0.07; HIV GKO/BaL, *p* = 0.23) (Appendix A). However, assuming a normal distribution (based on QQ plots) and performing a paired *t*-test for time points to measure the statistical difference linked to sex, we observed a significant enhanced GFP production in the male cells compared to the female cells (HIV GKO/VSV-G, *, *p* = 0.03; HIV GKO/LAI, **, *p* = 0.003; HIV GKO/BaL, **, *p* = 0.001), suggesting an increased HIV protein production (Appendix A). This was shown as well for older donor cells, except upon HIV GKO/VSV-G infection (HIV GKO/VSV-G, *p* = 0.76; HIV GKO/LAI, ** *p* = 0.001; HIV GKO/BaL, ** *p* = 0.002) (Appendix A). Finally, the AUC of the latent cell proportion over time p.-i. (as reflected by the proportion of mKO2+ GFP− cells over the total mKO2+ cell population) was significantly increased in younger donors upon infection with the three vectors (HIV GKO/VSV-G, ** *p* = 0.002; HIV GKO/LAI, ** *p* = 0.003; HIV GKO/BaL, ** *p* = 0.01) (Appendix A). Overall, these data are complementary to the infection success measured as the proportion of susceptible cells (mKO2+) or global permissiveness (mKO2+ GFP+).

Altogether, these data suggest that sex and age impact cell permissiveness to HIV infection in vitro, depending on the viral entry route. Globally, the male and older donor cells displayed an increased susceptibility and showed an increased permissiveness to HIV infection (VSV-G- and/or LAI-mediated entry), reflecting a more favorable environment for HIV entry, integration, and viral protein production. Consistently, the younger donor cells appeared to favor latent infection.

### 3.3. Activation-Induced Marker Expression Is Biased by Sex and Age and Correlates with Cell Permissiveness to HIV

Cellular activation is a pivotal parameter in cell permissiveness to HIV infection. To address whether cell activation features might explain the sex- and age-specific differences observed in cell permissiveness to HIV, we monitored the expression of selected activation markers (CD69, CD25, and HLA-DR) and ICIs (PD-1, CTLA-4, and TIM-3), as well as HIV entry receptors (CD4, CXCR4, and CCR5) over time. The proportion of cells expressing HIV entry receptors showed no sex-specific bias (CD4, *p* = 0.99; CXCR4, *p* = 0.95; and CCR5, *p* = 0.57) (Figure 3A), suggesting that they do not account for the observed differences. An analysis of activation markers showed that the male cells expressed significantly more CD69+ cells upon TCR-mediated stimulation than the female cells (***, *p* = 0.0007). However, CD25 and HLA-DR displayed no sex-specific bias (CD25, *p* = 0.25; HLA-DR, *p* = 0.77). Consistently, PD-1 and CTLA-4 were expressed on a significantly larger proportion of cells in men (PD-1, **, *p* = 0.009; CTLA-4, **, *p* = 0.009) and TIM-3 displayed a non-significant trend in the same direction (*p* = 0.05). Similarly, an age-differentiated analysis of the activation kinetics showed a significant, higher proportion of cells from older donors expressing CTLA-4 (*, *p* = 0.01) (Figure 3B). Importantly, CD4 and CXCR4 were present in a larger proportion of cells derived from the older donors (CD4, ***, *p* = 0.0002; CXCR4, *, *p* = 0.02), and it cannot be excluded that they might account for the HIV susceptibility’s difference, as they were expressed by >95% cells, resulting in a quasi-ubiquitous availability of HIV receptors. Similar to the infection data, we performed a paired *t*-test for each time point to measure the statistical difference linked to sex or age groups. We also found an increased expression of CD25 (*, *p* = 0.03) and TIM-3 (*, *p* = 0.02) in the male cells compared to the female cells, as well as CD25 (*, *p* = 0.02) and HLA-DR (***, *p* = 0.0002) in the older donor cells compared to the younger donor cells (Appendix A).

To assess whether the expression of these markers was linked to cell permissiveness to HIV, we correlated the relative expression of cell surface protein with relative GFP expression of HIV-infected cells over time p.-s. for all three HIV GKO viruses for each donor separately and found positive correlations with the activation markers and ICIs (Figure 3C). CTLA-4’s expression kinetics correlated well with the infection kinetics of HIV GKO/VSV-G (*p* < 0.05 in 20/20 donors, mean R-square = 0.86) and HIV GKO/LAI (*p* < 0.05 in 18/20 donors, mean R-square = 0.74), whereas the correlation was only partial with HIV GKO/BaL (*p* < 0.05 in 12/20 donors, mean R-square = 0.75). PD-1’s relative expression displayed a high correlation with infection through LAI-mediated entry (*p* < 0.05 in 17/20 donors, mean R-square = 0.80) and a milder one through VSV-G-mediated entry (*p* < 0.05 in 10/20 donors, mean R-square = 0.65). CD69 was only found to exhibit a significant correlation with virus infection using BaL-mediated entry (*p* < 0.05 in 18/20 donors, mean R-square = 0.86), potentially highlighting a higher dependence of R5-tropic viruses to cells in early activation.

In summary, these data identified sex- and age-specific differences on activation kinetics, as revealed through the expression of selected activation markers and ICIs. In particular, the cells from men and older donors showed a higher cell permissiveness and higher marker expression levels. Moreover, we highlighted that the relative expression kinetics of CD69, PD-1, and CTLA-4 correlated with the permissiveness to HIV on an entry-dependent mode.

### 3.4. CD4+ T Cell Pool Composition Is Not Biased by Sex

To address whether sex bias in permissiveness to HIV could be linked to CD4+ T cell subpopulation composition, we stimulated cells from two female and two male donors for 24 h and performed single-cell-RNA-Seq (sc-RNA-Seq). We investigated a total of 24,626 single cells and mapped them against three different references (i.e., DICE, Monaco, and Seurat [25,27,28]) (Figure 4A, Appendix A). As each database was constructed using different experimental protocols and addressed different layers of CD4+ T cell subpopulations, this process allowed us to have a comprehensive understanding of the subpopulation distribution at the subtype and differentiation stage levels. The DICE annotation did not identify any sex- specific subpopulation enrichment and revealed that naïve stimulated T cells represented more than 65% of the cells in both sexes (Figure 4B). Other cell types, including memory T regulatory (Treg), naïve, naïve Treg, Tfh, Th1, Th1/Th17, Th17, and Th2, accounted only for minor cell fractions. Consistently, the annotation with the Monaco reference did not reveal any sex bias either. This latter dataset showed an enrichment of Treg and Th1/Th17, followed by naïve and Th1. The remaining cells were split between Tfh, T terminal effector (Tte), Th17, and Th2. Finally, the Seurat annotation showed a comparable enrichment of CD4+ T cell subpopulations between men and women, with T central memory (Tcm) representing more than 70% of the cells in both, and the rest being divided between cytotoxic T lymphocytes (CTL), naïve, proliferating, T effector memory (Tem), and Treg.

Overall, the comparison of our four donors with the three reference databases and the twenty-three annotated subtypes did not reveal sex-specific cell subtype distribution, suggesting that the differential permissiveness to HIV between male and female cells cannot be explained with cellular subpopulation composition.

### 3.5. Identification of Sex-Specific Gene Expression and of Sex and Age Impact on Activation-Linked Pathways

To further investigate and identify the cellular determinants responsible for donor-related permissiveness to HIV, we stimulated CD4+ T cells from each of the twenty donors for 24 or 72 h and performed bulk transcriptomic analyses. The principal component analysis (PCA) revealed that 71% of the variance between the samples was explained through activation, as expected, while 4% was attributed to sex (Figure 5A). Over a total of 13,154 detected genes, the differential gene expression analysis (DGEA) between men and women cells at 24 h p.-s. outlined 28 differentially expressed genes (DEG). The top DEG genes were mostly linked to X and Y chromosomes, with X-linked genes enriched in women and Y-linked genes enriched in men and absent in women, as expected (Figure 5B). To ensure a balanced biological function between the sexes, most X- and Y-linked genes unrelated to reproductive functions harbor a corresponding paralog on the other chromosome. As most of the DEG between men and women are linked to sex chromosomes, we interrogated the combined expression of corresponding paralogs in both sexes (Figure 5C). At 24 h p.-s., five out of the nine gene pairs presented a differential average expression in function of sex, with *EIF1AY/EIF1AX* (*p* = 1.4 × 10^−10^), *RPS4Y1/RPS4X* (*p* = 6.3 × 10^−6^), *DDX3Y/DDX3X* (*p* = 7.8 × 10^−6^), and *PRKY/PRKX* (*p* = 0.0065) enriched in women, while *USP9Y/USP9X* (*p* = 2.9 × 10^−7^) enriched in men. The translation of differential gene expression on functional pathway regulation was then assessed using a hallmark gene set enrichment analysis (GSEA). To further pinpoint how activation affects gene expression and functional pathway enrichment, the 72 h and 24 h stimulations were compared. The assessment of the stimulation effect outlined 5266 and 5220 DEG when comparing 72 h to 24 h stimulation in women and men, respectively (Appendix A), and showed functional enrichment of pathways related to the cell cycle (e.g., E2F targets or G2M checkpoint) and metabolism (glycolysis, oxidative phosphorylation, or mTORC1 signaling), demonstrating the upregulation of genes linked to activation, together with a downregulation of innate immunity (Appendix A). Similar results were obtained in men and women (Appendix A). A further interrogation of all the donors by sex did not result in any significantly enriched pathway. As this may be the result of age-dependent variation, functional analyses of sex-induced effect were evaluated within age classes. Comparing the sexes within the <50 years and ≥50 years age categories did not result in any enrichment. As we reasoned that reproductive hormone status might influence the data, we further divided the donors into subcategories (i.e., 22–35, 36–52, and 53–72 years). Thus, we observed a variable effect only in the 36–52 years old donors, as it is probably in these donors that the most hormonal changes occur as it is the age group where women are most prone to enter menopause. The DGEA resulted in 156 genes, enriching pathways in men compared to women at 24 h p.-s. related to the activation or cell cycle as MYC targets, mTORC1 signaling, oxidative phosphorylation, glycolysis, or IL-2/STAT5 signaling, while IFN-α’s response was enriched in women (Appendix A). These data suggest that activation is strengthened in men compared to women in age group 36–52 years.

Functional pathway regulation was further investigated at the age level and enrichment of activation-linked pathways such as oxidative phosphorylation, mTORC1 signaling, or IL2 STAT5 signaling was evidenced in the older donors compared to the younger ones at 24 h p.-s. (Appendix A). When analyzing only the women’s samples, the effect was more pronounced as older women showed an enrichment of MYC targets, glycolysis, mTORC1 signaling, PI3K Akt mTOR signaling, G2M checkpoint, or E2F targets (Figure 5D). Our results support the conclusion that activation potency is stronger with age and appears more pronounced in women.

Altogether, these results suggest that sex and age influenced the cellular transcriptome of CD4+ T cells. We evidenced biased, combined expression of paralogs on X and Y chromosomes by sex, as well as an enrichment of activation-linked pathways in donor categories most susceptible to HIV infection, i.e., men (age-dependent) and donors ≥ 50 years old, suggesting that an enhanced permissiveness to HIV is related to an enhanced activation potency.

## 4. Discussion

Permissiveness to HIV infection has been known to differ between individuals for more than two decades [39]. However, most efforts focused on identifying cell features involved in permissiveness per se, but lacked individual-related considerations, such as sex or age [40]. In this study, we hypothesized that both factors could impact the permissiveness to HIV in a CD4+ T cell activation kinetics. We thus first tracked the activation levels and the permissiveness phenotype differentially, according to sex or age. We further deciphered sex- and age-specific transcriptomic landscapes to identify genes correlating with cell permissiveness to HIV.

We isolated and stimulated primary CD4+ T cells from twenty HIV-negative blood donors over a six-day (144 h) period and infected them with HIV GKO dual-reporter (EF-1α-mKO2 and LTR-GFP) pseudotyped with VSV-G (amphotropic), BaL (R5-tropic), or LAI (X4-tropic) envelopes. Consistent with previous reports, the HIV infection success increased with time p.-s. [41]. In our experimental system, the cells were most permissive between 24 and 72 h p.-s., and the kinetics were dependent on the viral envelope. Viral entry upon VSV-G pseudotyping was not limited by the expression of HIV receptors and, therefore, reflects the permissiveness dictated by the intracellular environment, while the use of BaL and LAI envelopes also relied on the expression of HIV entry receptors. Our data showed that cell susceptibility kinetics to HIV GKO/LAI was similar to the HIV GKO/VSV-G’s, which suggests that the receptors exploited upon LAI-mediated entry were not limiting infection. Conversely, the susceptibility kinetics to HIV GKO/BaL appeared to be shifted, happening one day earlier, suggesting that CCR5’s low levels were a limiting factor, impairing efficient infection. Importantly, the vectors harboring the HIV envelopes displayed reduced infection rates compared to the vector harboring the VSV-G envelope. In addition, HIV GKO lacks Nef, resulting in a decreased uncoating efficiency upon infection with native HIV envelopes but not with VSV-G [42]. Of note, the presence of a *ccr5*Δ*32* mutation may lead to a reduced susceptibility to an R5-tropic vector. Although *ccr5* was not genotyped, CCR5 expression was detected in all the samples, indicating that no donor was homozygous for *ccr5*Δ*32* mutation [43]. However, our data cannot exclude heterozygous cells that would express lower amounts of CCR5 and, thus, restrict R5-tropic vector susceptibility.

Upon analysis of infection efficiency by sex, we showed that the CD4+ T cells derived from men displayed an increased susceptibility to HIV within the most permissive 24–72 h time window. These results are concordant with a previous analysis performed ex vivo on cells derived from people living with HIV, evidencing a reduced HIV RNA synthesis in cells from women [12]. For the first time, our study enabled to recapitulate these findings in an in vitro infection model, suggesting that it can be the result of intrinsic cellular composition, without immune mediation, at least in part. As the size of the latent reservoir was estimated to be comparable between men and women, this would imply an enhanced HIV-mediated cytotoxicity of the infected cells in men [12]. Our work evidenced a sex bias upon HIV infection mediated with a VSV-G or LAI envelope, but not BaL. This could likely be explained by the reduced infection success using HIV GKO/BaL (1.1% on average in peak infection). However, when we looked at the stability of the fluorescent reporters, we showed that the frequency of GFP+ cells over time p.-i. was significantly higher in the male cells for the three entry modes, further supporting our hypothesis. Previous results from Bosque and Planelles’ groups on memory CD4+ T cells reported the absence of a sex-specific impact on the susceptibility to HIV [44,45]. This suggests that the sex difference identified in the present study is unlikely to have been caused by a biased memory cell fraction.

Given the extent of immune-related changes in later life, it was possible to envision age-related implications on HIV infection. Yet, studies on a direct impact of age on cellular susceptibility to HIV are still lacking, as only one work reported a negative correlation between age and HIV infection success in memory CD4+ T cells from women [45]. Our analysis outlined that the donors aged 50 years or older displayed an increased susceptibility to HIV infection in the most permissive 24 h–72 h time window, upon LAI-mediated HIV entry. As previously mentioned, the low infection efficiency did not allow the identification of a significant impact in the BaL setting. Similar to the sex analysis, the controlled expression of fluorescent reporters over time p.-i. resulted in significantly higher levels in the older donor category upon entry mediated using both HIV envelopes. However, no effect of age was observed upon VSV-G entry, suggesting that age-dependent differences are related to a gp120-induced signaling cascade.

Upon the kinetics of TCR-mediated stimulation, we found a significantly higher proportion of CD4+ T cells from men compared to women expressing CD69, PD-1, and CTLA-4, as well as a trend for CD25 and TIM-3, all of which are surface markers upregulated upon activation. Similarly, the proportion of positive cells was significantly higher for CTLA-4 and exhibited a trend for CD25 and HLA-DR in donors aged 50 years or more compared to younger ones. These results suggest that the higher permissiveness of cells derived from men or older donors is notably due to a higher activation. Indeed, we showed that, as activation progressed, innate immunity is downregulated and transcriptional activity is increased, likely explaining the increasing HIV permissiveness. Of note, a previous study found contrasting results regarding sex bias in activation, with female CD4+ T cells exhibiting higher CD69 expression [46]. This outcome may originate from the different stimulation methods. Indeed, the phytohemagglutinin (PHA) used in the previous study engender a different signal transduction by crosslinking CD3 without inducing CD28 co-stimulation, as performed in our system, which can alter intracellular signaling and, therefore, result in a different outcome. As CD28 co-stimulation is necessary in vivo to induce a successful activation, the use of αCD3/CD28 in the presence of IL-2 in our study might, thus, be more representative of a physiological stimulation. In addition, we found positive correlations between the kinetics of CD69, CTLA-4, and PD-1 relative expression and cell permissiveness to HIV, with at least one of the three viral reporters used. Of note, CTLA-4 and PD-1 were reported to be enriched on the surface of latently infected cells [47,48]. Our results suggest that these markers are expressed at the moment of infection and are not upregulated as the result of HIV-induced changes.

The single-cell analysis with 23 subtype annotations of CD4+ T cell pool composition exhibited no sex-specific enrichment of any subpopulation, which suggests that the difference in permissiveness to HIV is not attributed to a specific subtype but rather to a cellular state. While previous studies evidenced higher levels of Th1 and Th2 in women and higher Th17 and Treg in men, our results did not recapitulate these findings [5]. However, this analysis may necessitate an increased number of donors in order to reveal a statistically significant tendency. This could also be the result of the stimulation method used, i.e., TCR-mediated in the presence of IL-2, which affects the subtype’s composition as compared to the antigen–antigen presenting cell complex [49]. A switch of the stimulation method might, thus, favor the differentiation of specific subtypes and reveal a sex bias in a sc-RNA-Seq. Furthermore, we detected Tfh in the peripheral blood, although not abundantly. However, as they were not isolated directly from the germinal centers, where the majority of Tfh is located, the resemblance between the germinal center Tfh and the circulating Tfh might be limited. Therefore, cells from lymph nodes’ biopsies would likely be more appropriate for the specific study of Tfh.

The transcriptomic analyses outlined a differential sex and age regulation in the CD4+ T cells prior to the exposure to HIV, with DEG linked to sex chromosomes in men compared to women. To potentially compensate for the absence of X- or Y-linked genes, most of the genes identified in the present study harbored a paralog displaying a conserved function on the other sex chromosome [50]. The combined expression of both paralogs showed a sex bias in the average expression, with increased levels of *EIF1AY/EIF1AX*, *DDX3Y/DDX3X*, *RPS4Y1/RPS4X*, and *PRKY/PRKX* in women. Interestingly, EIF1A, DDX3, and PRKX were demonstrated to enhance HIV replication, suggesting that their activity is not sufficient to counter the sex bias in HIV susceptibility [51,52,53]. Previous studies identified RPS4 as interfering with RRE in HIV and as associated with a decreased hepatitis C virus’ replication, making it a potential HIV inhibitory factor [54,55]. Conversely, the *USP9Y/USP9X* levels were increased in men. A facilitating role was shown in the gammaherpesvirus, but a putative interaction with HIV remains to be deciphered [56]. Our transcriptomic data also showed that donor categories displaying an enhanced HIV susceptibility presented enriched cellular pathways linked to the activation and cell cycle, which might explain the higher permissiveness to HIV. Of note, upon the sex comparison, we found an enrichment of these pathways only upon further splitting of the age categories, in the 36–52 years old donors. This suggests that the sex bias in the activation is linked to the reproductive period in women, with hormone exposure participating in reducing activation in women, no longer impacting it after menopause. Indeed, this hypothesis is supported by the fact that progesterone was recently shown to dampen activation in CD4+ T cells [57]. The absence of evidence upon the comparison of donors younger and older than 50 years is, therefore, probably the result of differential hormonal regulation that can be influenced by the state of the hormonal cycle, the use of hormone-based contraceptive, and a recent pregnancy. Future studies should consider steroid hormones in the role of cellular activation and permissiveness to HIV.

Our study has several limitations that have to be considered for data interpretation. First of all, our analysis was performed on twenty donors, which might limit statistical power. Second, a higher probability of HIV seroconversion in women through heterosexual intercourse cannot be explained with cellular susceptibility [8,9], as our data showed that cells from male individuals were more susceptible in vitro than female cells. Thus, hormone exposure and local inflammation are more likely to affect the mucosa barrier and render it more permissive to HIV [18]. Furthermore, circulating cells might not entirely recapitulate the phenotype of tissue cells. Finally, there are many confounding factors that could also impact cell susceptibility to HIV that were not taken into account in this study due to anonymous blood donations, for which only limited information was available. These include ethnical ancestry, comorbidities, cytomegalovirus seropositivity, or hormonal status. These factors may be considered in future studies including more donors.

In conclusion, this study outlined the importance of considering sex and age of donor cells in HIV infection studies and identified novel genes potentially impacting cell permissiveness. Future work should further confirm the role of these genes in HIV replication and identify the potential links between individual activation potency and HIV latency reactivation. Indeed, deciphering reactivation mechanisms might help improve the spectrum of latency reactivation strategies and, thus, purging the reservoir.

## Figures and Tables

**Figure 1 cells-12-02689-f001:**
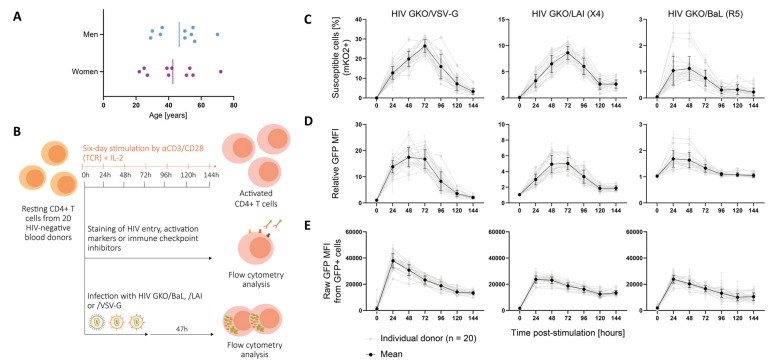
Cell permissiveness to HIV evolves with cell activation dynamics. (**A**) Panel of twenty HIV-negative blood donors included ten men and ten women of age ranging from 22 to 72 years. The vertical bar represents the mean age for men and women, respectively. (**B**) Resting CD4+ T cells from 20 HIV-negative blood donors were stimulated with αCD3/CD28 in presence of IL-2 over a 144 h period. After each 24 h period, the cells were collected and were either stained for cell surface proteins or infected with the HIV GKO dual-reporter vector. The activation state was tracked by measuring the cell surface protein’s expression of activation and ICIs, as well as HIV entry receptors, using fluorophore-conjugated antibody staining. Each staining was performed in duplicates and assessed using flow cytometry. In parallel, cells were infected with HIV GKO harboring distinct viral envelopes (VSV-G, X4-tropic LAI, R5-tropic BaL) in biological duplicates. Cell permissiveness to HIV was assessed 47 h p.-i. using flow cytometry. (**C**–**E**) Infection success over time p.-s. for the three HIV GKO vectors harboring distinctive envelopes (VSV-G, LAI, BaL). The grey lines represent the infection kinetics obtained for each individual donor and are the mean of the biological duplicates. The black line represents the mean of all the donors. The error bars represent the SD. (**C**) The proportion of cells susceptible to HIV infection was assessed with cells expressing EF1α-driven mKO2. (**D**) The global cell population permissiveness to HIV was assessed as a relative GFP MFI, normalized by the corresponding mock-infected control, informing the success of entry, integration, and productive LTR-driven GFP expression. (**E**) The intracellular permissiveness level was assessed by measuring the raw GFP MFI in GFP+ cells.

**Figure 2 cells-12-02689-f002:**
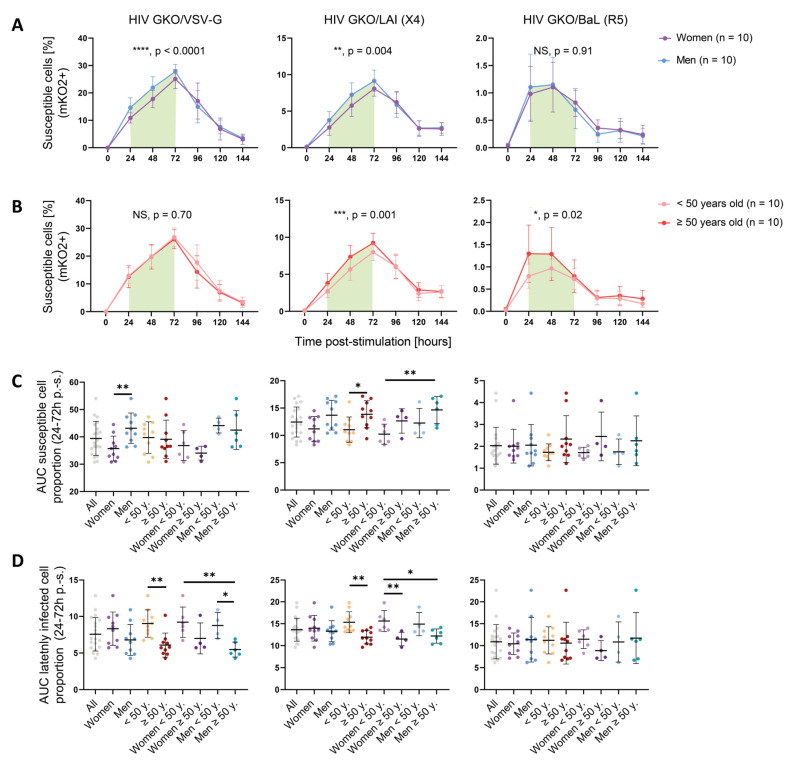
Sex and age impact CD4+ T cells susceptibility to HIV infection. (**A**,**B**) Proportion of susceptible cells (mKO2+) over time p.-s. for the three HIV GKO reporters (VSV-G, LAI (X4), BaL (R5)), separated according to sex (women: purple; men: blue) (**A**) or according to age (<50 years old: pale orange; ≥50 years old: red) (**B**). The lines represent the mean of each category per time point. The error bars represent the SD. Statistical differences were calculated between the donor groups during the 24 and 72 h p.-s. time window (indicated by the green box) using two-way ANOVA. NS: not significant (*p* > 0.05). * *p* < 0.05, ** *p* < 0.01, *** *p* < 0.001, **** *p* < 0.0001. (**C**,**D**) Proportion of susceptible cells (mKO2+) (**C**) or latently infected cells (mKO2+ GFP− over mKO2+ population) (**D**) to HIV, calculated as the AUC between 24 and 72 h p.-s. and according to multiple donor categories. The infection was performed with the three HIV GKO vectors (VSV-G, LAI (X4), BaL (R5)). Each dot represents the mean of the biological duplicates of one donor. Statistical differences between sex and age were calculated using two-way ANOVA. * *p* < 0.05, ** *p* < 0.01.

**Figure 3 cells-12-02689-f003:**
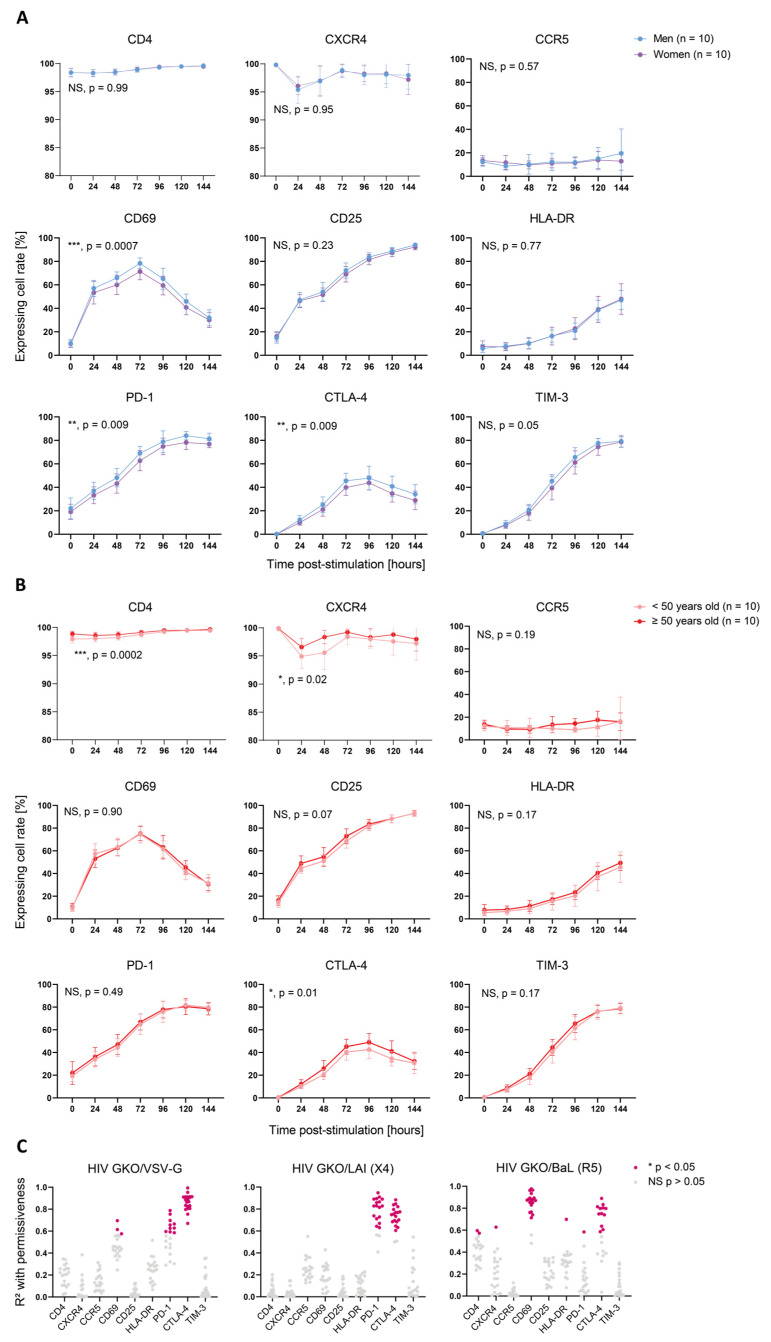
Activation-induced marker expression is biased by sex and age and correlates with cell permissiveness to HIV. (**A**,**B**) Proportion of cells expressing selected cell surface proteins over time p.-s. The proteins analyzed include HIV entry markers (CD4, CXCR4, and CCR54; upper graphs), activation markers (CD69, CD25, and HLA-DR; middle graphs), and ICIs (PD-1, CTLA-4, and TIM-3; lower graphs). The lines represent the mean of each category per time point. The error bars represent the SD. The statistical analysis between the donor groups was performed using two-way ANOVA. NS: not significant (*p* > 0.05). * *p* < 0.05, ** *p* < 0.01, *** *p* < 0.001. (**A**) Surface maker expression according to sex. Women are represented in purple and men in blue. (**B**) Surface marker expression according to age. The donors younger than 50 years old are represented in pale orange and the donors 50 years old or older in red. (**C**) Correlation analysis between protein surface expression (MFI of stained sample normalized to corresponding non-stained sample) and permissiveness to HIV (relative GFP MFI, normalized to corresponding mock-infected sample) for the three HIV GKO vectors throughout activation kinetics, as follows: (**top**) VSV-G; (**middle**) LAI (X4); and (**bottom**) BaL (R5). Each dot represents the R-square value calculated using linear regression for one donor with the mean of infection duplicates and the staining duplicates. Non-significant (NS) correlations (*p* > 0.05) are represented in grey and significant ones (*, *p* < 0.05) in pink.

**Figure 4 cells-12-02689-f004:**
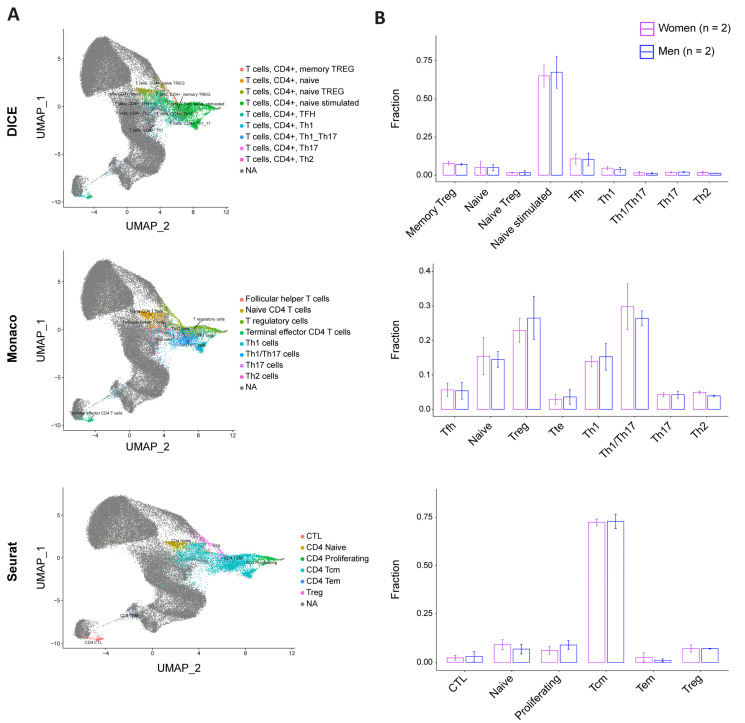
CD4+ T cell pool composition is not biased by sex. Single-cell distribution was assessed using three reference datasets: DICE (**top**), Monaco (**middle**), and Seurat (**bottom**). (**A**) UMAP projections of our four-donor cell distribution pattern (colored) on reference datasets (grey). (**B**) Cell subtype fraction by sex and according to the corresponding database annotation ((**top**): DICE; (**middle**): Monaco; and (**bottom**): Seurat). Women are represented in purple and men in blue. The error bars represent the SD. The statistics were calculated using a *t*-test, which reported no significant *p*-value for all the comparisons.

**Figure 5 cells-12-02689-f005:**
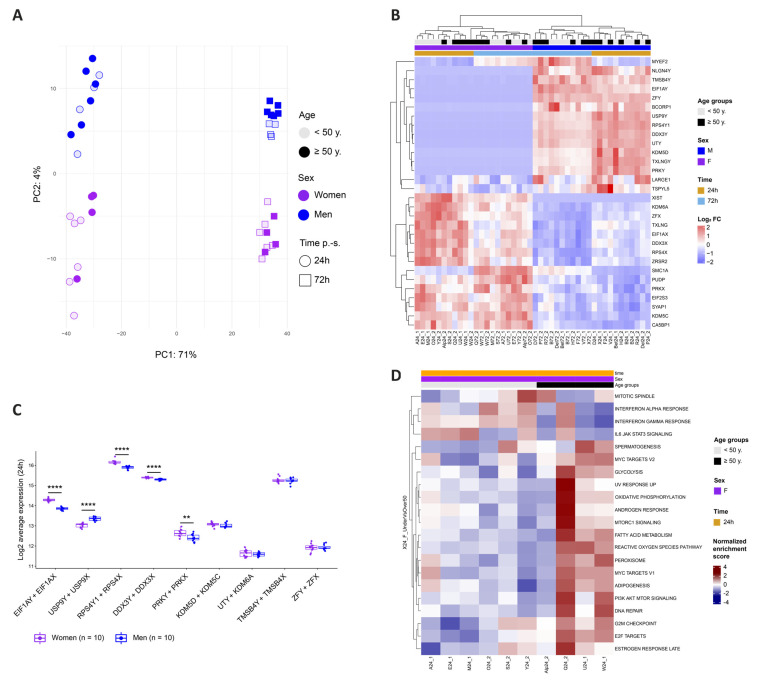
Identification of Sex-Specific Gene Expression and of Sex and Age Impact on Activation-Linked Pathways (**A**) PCA plot with explained percentage of variance between cell sample transcriptomes. CD4 + T cells were stimulated for 24 h (circles) or 72 h (squares). The women’s samples are depicted in purple and the men’s samples in blue. Age is represented with color intensity (<50 years in a light shade and ≥50 years in a dark shade). (**B**) The top differentially expressed genes between men’s and women’s cells at 24 h p.-s. The log2 FC of each gene compared to its average expression is represented in shades ranging from red to blue. (**C**) Combined expression at 24 h p.-s. of Y-linked genes and their X chromosome paralogs in men (in blue) and in women (in purple). The statistical differences were calculated using a *t*-test. **** *p* < 0.0001, ** *p* < 0.01. (**D**) Significantly enriched hallmark gene sets between women < 50 years and women ≥ 50 years at 24 h p.-s. The normalized enrichment score is represented in shading ranging from red to blue.

## Data Availability

All the transcriptomic data discussed in this publication have been deposited in the NCBI’s Gene Expression Omnibus [58] and are accessible through the GEO Series accession number GSE247486 (https://www.ncbi.nlm.nih.gov/geo/query/acc.cgi?acc=GSE247486). All the flow cytometry data are openly accessible in the Zenodo repository (https://doi.org/10.5281/zenodo.10091514).

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
