# Peer review of "Sex and Age Impact CD4+ T Cell Susceptibility to HIV In Vitro through Cell Activation Dynamics"

_cells, 2023, doi:10.3390/cells12232689_

Round 1
Reviewer 1 Report
Comments and Suggestions for Authors
This submitted manuscript by Ludivine Brandt et al. explores the impact of sex and age on CD4+ T cell susceptibility to HIV infection in vitro. Using pseudovirus infection studies of CD4 T cells isolated from cryopreserved peripheral blood mononuclear cells, the study finds that cells from men and older participants (over 50 years old) exhibit increased susceptibility to HIV infection, primarily between 24 and 72 hours post-stimulation, compared to CD4 T cells from women and participants under 50 years old. This increased susceptibility correlates with higher expression of activation markers and immune checkpoint inhibitors, suggesting enhanced activation in these groups. Transcriptomic analysis reveals differences in gene expression patterns related to activation and cell cycle in male and older donor cells. Overall, this study highlights the role of sex and age in influencing cellular permissiveness to HIV infection, emphasizing the importance of understanding individual factors in HIV research.
The impact of this paper lies in its contribution to our understanding of how sex and age influence the susceptibility of CD4+ T cells to HIV infection. It provides insights into the cellular mechanisms that affect viral replication, particularly during the early stages of activation. This knowledge from this paper can inform future studies on HIV cure strategies and improve patient care, considering the variations in viral replication at the cellular level. This study is descriptive, low in sample size, the in vitro infection model is artificial, and there are several issues that should be addressed:
1. The major limitation of this study is sample size. The overall study is of 20 individuals and the transcriptomic analysis only uses 4 individuals (2 male and 2 female). This needs to be brought out in the discussion. There are other factors that could also influence the results such as host genetics, environmental factors, comorbidities, etc. Is there any information on the donors other than age and sex? Are “HIV-negative” donors from Switzerland and how does this relate to Sub-Saharan Africa where most infections in women occur? Are any of these donors CCR5 delta 32 hetero or homozygous? What other infections or co-morbid conditions were observed in the study participants?
2. The study uses cryopreserved PBMC to study the impact of sex and age on susceptibility to HIV infection. Was viability checked after thaw and was there any variability in the quality of cells across donors which could impact in vitro HIV infection? Was a viability marker included in the flow analysis to exclude dead cells? How efficient was the negative selection for the CD4 T cells? Was there variability in the purity of CD4 T cells after negative isolation?
3. Insufficient methods were provided concerning the flow cytometry. How was the optimal titer of antibodies used in the experiments? How were gates determined for the phenotypic analysis? Was field minus one condition used to establish gates? Example gating for the phenotypic markers should be included in the supplement. How was optimal staining determined for the antibodies? Tess Brodie showed that chemokine receptor staining was sensitive to several experimental variables including staining temperature (PMID 23504907). Was the optimal temperature considered or tested here?
4. Are the authors able to use combinatorial activation and inhibitory marker expression for the analysis? It would be interesting to see if CD25+HLADR+ CD4 T cells, CD25+PD1+, or other combinations of activation markers and checkpoint inhibitors to see if you get better resolution or correlations with infection.
5. Some of the receptor expression is very high after stimulation. How relevant is this to in vivo infection? Most infections in women occur at the mucosal surface. How does this data relate to mucosal infection? In line with this, the single-cell RNA-seq identifies populations of CD4 T cells that are not abundant in peripheral blood like Th17 and Folicular helper cells. By definition, bona fide Follicular helper cells should be in other compartments (and the significance of their peripheral blood counterparts is not understood). This limitation could be noted in the discussion.
6. It is unclear if the bulk cell transcriptomics was from the 4 donors used in the single cell RNA-Seq work or the entire 20 donors in the infection and flow cytometry experiments. Please clarify.
Author Response
Dear Editor,
Dear Reviewers,
We thank you for the opportunity to revise our work and thank the reviewers for their valuable comments. We have now answered all the queries (please see the attachment) and included additional supporting materials to provide appropriate response. We hope that the revised version of the manuscript will meet your expectations and that you will find it appropriate for publication.
Sincerely,
Angela Ciuffi and Ludivine Brandt

Reviewer 2 Report
Comments and Suggestions for Authors
Brief summary:
The study presented in the manuscript by Brandt et al. aimed to investigate the impact of sex and age to the susceptibility of activated CD4+ T cells to infection with HIV in vitro. The rationale of performing these experiments in vitro was to investigate sex- and age-based differences without hormonal influences and biases caused by a prolonged HIV infection. The authors demonstrated greater susceptibility to HIV infection of cells isolated from males and from donors >50 years old. Furthermore, susceptibility to HIV infection correlated with the cellular activation/exhaustion phenotypes. The strengths of this study are a large sample size (N=20), experimental and analytical rigor, and clear logic of the experiment presented.
General concept comments:
1). Statistical analyses are not clear. For graphs such as in Figure 2A,B, only one p-value is shown, with the statement in the methods and in the figure legend that a paired t-test was used. However, the graphs show 3 time points when men and women or old and young participants are compared. This design implies that there should be some 2-way test performed. Alternatively, comparisons can be conducted at different time points separately (e.g. men vs women), but this test would not be paired, since samples come from different people. Furthermore, the test is conducted on proportions of cells, therefore, normality cannot be assumed and some non-parametric test (not a t-test) should be used instead.
2). Subdividing donors into age categories for analysis of sex differences appears to be superficial. In lines 493-495, the authors state that “reproductive hormone status might influence the data”, so the donors were separated into categories by age, 22-35, 36-52 and 53-72. This division seems to be arbitrary. In women, it makes sense to break the samples into pre- and post-menopausal states, which would more or less be consistent with the original age groups that the authors compared (<50 and >50 years old). It is not expected that hormonal differences between 22-35 and 36-52 would be as robust as the differences before and after 50 years old. If subsets of samples were selected at random from the 22 to 52 years old group, how frequently would the observation hold that there are differences between men and women? Ideally, bootstrapping analysis should be done to ensure that the observed differences between men and women in the age group of 36-52 years old are not a random result. Another important thing to note in the manuscript, when talking about hormone influence, is whether all the women were not on hormonal contraceptives. This is not mentioned anywhere in the manuscript.
Specific comments:
1). Methods: section “Population RNA-Seq library preparation and sequencing”, were these mRNA libraries, or ribo-depleted libraries?
2). Data availability: single-cell RNA-Seq and population RNA-Seq data should be deposited into a public repository and an accession number should be provided in the manuscript. If this is not done, readers will not know where to look for the data when the manuscript is published. Gene Expression Omnibus repository generates an accession number and reviewer access token (without public access) for the time papers are under consideration.
3). Graphs, where percentages of cells with expression of some markers nearing 100%: it is hard to see any differences in expression at such a scale. Perhaps, these graphs should have some lower limit, e.g. 80%, rather than 0, so that the differences could be seen better. Examples: Figure 3, Figure S2, Figure S3.
4). Figure S3D: besides being hard to see a decrease in productively infected cells, it appears that at least for VSV-G and R5 envelopes the proportions of latently infected cells are increasing. Can it be concluded that the decrease of productively infected cells in this case is due to cell death, not due to latency establishment (line 284)?
5). Lines 457-458: please check this statement. On Figure 5A, it appears that samples from men at 72 hours p.s. clustered by age.
6). Lines 484-489: it is not clear what the rationale was to make comparisons between 24 h and 72 h post stimulation. Stimulation signal would likely prevail over any other differences between the samples, so it is not surprising that pathways related to cell cycle were identified, and were the same for women and men. A much more informative comparison would be between women and men at each of the time points. If the authors choose to keep inclusion of analysis between the time points, analysis for men should also be presented, and perhaps a Venn diagram of pathways identified for women and men should be added.
7). Line 557: please clarify the meaning of “imputed”.
Comments on the Quality of English LanguageMinor editing is needed, such as verb and noun agreements, e.g. line 31 "cellular population HAS to express"; line 617 "this suggestS". Please check the entire manuscript.
Author Response

(The authors gave the same response as above.)

Round 2
Reviewer 1 Report
Comments and Suggestions for Authors
The authors have satisfactorily addressed the majority of comments, however, two comments remain:
1. For initial comment #3 concerning flow methods and antibody titration. Using an arbitrary volume of monoclonal antibody for staining is not optimal. It remains unclear if the authors were following the manufacturer's instruction ("5 µL per million cells in 100 µL staining volume") since the authors stained 50,000 cells and used 0.5ul and undefined volume. With that being said, the data presented in this paper is most likely not impacted. The authors should provide example staining of the phenotypic markers in the supplemental material to provide assurance the staining is adequate. In the future, the authors should use optimal titers of antibodies to ensure reproducibility of flow cytometry results. There are a number of publications on this such as PMID: 31577065.
2. In regard to the original question (#4) about combinatorial flow analysis, the authors state that to do this "would have also required many staining controls and more complicated flow cytometry data". Were multiple stains used in each panel or were the surface marker expression determined using single fluorescent parameter flow cytometry? If multiple monoclonal antibodies were used in each panel, how was compensation determined? The panels should be listed in the supplemental material.
Author Response
Dear Editor,
Dear Reviewer,
We thank you for this second opportunity to revise our work and thank the reviewers for their valuable comments. We have now answered all the queries (please see the attachment). We modified the manuscript to gain clarity and included additional supporting materials to provide appropriate response. We hope that the revised version of the manuscript will meet your expectations and that you will find it appropriate for publication.
Sincerely,
Angela Ciuffi and Ludivine Brandt
